# Diagonal Graph Convolutional Networks with Adaptive Neighborhood Aggregation

## Abstract

Graph convolutional networks (GCNs) and their variants have generalized deep learning methods into non-Euclidean graph data, bringing a substantial improvement on many graph mining tasks. In this paper, we revisit the mathematical foundation of GCNs and study how to extend their representation capacity. We discover that their performance can be improved with an adaptive neighborhood aggregation step. The core idea is to adaptively scale the output signal for each node and automatically train a suitable nonlinear encoder for the input signal. In this work, we present a new method named Diagonal Graph Convolutional Networks (DiagGCN) based on this idea. Importantly, one of the adaptive aggregation techniques—the permutations of diagonal matrices—used in DiagGCN offers a flexible framework to design GCNs and in fact, some of the most expressive GCNs, e.g., the graph attention network, can be reformulated as a particular instance of our model. Standard experiments on open graph benchmarks show that our proposed framework can consistently improve the graph classification accuracy when compared to state-of-the-art baselines.

## 1 Introduction

The past decade has witnessed the remarkable success of deep neural networks across different domains, ranging from computer vision (Krizhevsky et al., 2012; He et al., 2016), to speech recognition (Abdel-Hamid et al., 2014), to natural language processing (Gehring et al., 2017). These breakthroughs have provoked the interest to extend neural operations from linear or spatial domains to arbitrary graph structures, e.g., social networks, gene-protein networks, and knowledge graphs.

One challenge in learning with graph-structured data is how to represent each node or subgraph in neural networks. This is challenging, as graph data lies in the non-Euclidean space with complex structures and random size, making it infeasible to directly apply traditional neural network algorithms (Hamilton et al., 2017; Kipf & Welling, 2016). Recently, there have been attempts to generalize neural networks for handling graph-structured data. For example, graph neural networks (GNNs) are introduced as recursive neural networks in which node features are propagated iteratively until equilibrium (Gori et al., 2005; Scarselli et al., 2009).

More recently, several studies show that it is natural for convolutional neural networks (CNNs) to model graph data structure by defining spatial localized convolutional filters over graph Laplacian spectral space (Bruna et al., 2013; Henaff et al., 2015; Defferrard et al., 2016) or more directly, over graph neighborhoods (Duvenaud et al., 2015; Niepert et al., 2016; Monti et al., 2017). These techniques, named as graph convolutional networks (GCNs) (Kipf & Welling, 2016), enable deep neural networks to effectively capture graph topology and properties. Over the past three years, various GCN based models, such as GraphSAGE (Hamilton et al., 2017), FastGCN (Chen et al., 2018), and GAT (Velickovic et al., 2018), have been rapidly developed, offering promising results for important graph mining tasks, such as, node classification and link prediction.

In theory, GCNs can be considered as a simplification of the traditional graph spectral methods (Kipf & Welling, 2016). The common strategy is to model a node's neighborhood as the receptive field and then define the "graph convolution layer" as a mean pooling aggregation followed by a nonlinearity encoder. Existing GCNs stack multiple layers and learn node representations by recursively aggregating information from neighbors, and the analysis of their representational capacity can be found in a recent study (Xu et al., 2019). Essentially, the recursive convolutional aggregation process

in GCNs, or the Neural Message Passing framework (Gilmer et al., 2017), is also closely related to the Weisfeiler-Lehman (WL) graph isomorphism test (Weisfeiler & Lehman, 1968).

Though GCNs have been extensively explored, how to better capture the graph structures through convolutions is still largely an open question. Particularly, the neighborhood aggregation step in GCNs can be seen as the diffusion process in real-world networks (Gilmer et al., 2017), in which different nodes play different roles in the propagation. For example, for information diffusion in social networks, opinion leaders (Granovetter, 1973) are more likely to expose information to associated communities and structural holes (Burt, 2009) can spread information across different groups. However, the neighborhood aggregation step in most existing GCNs considers neither the different structural importance of nodes within each step nor the dynamics of node importance across different propagation steps (Kipf & Welling, 2016; Hamilton et al., 2017; Chen et al., 2018).

In light of this limitation, we propose an adaptive neighborhood aggregation strategy to model and aggregate the dynamics of node importance. The adaptive neighborhood aggregation step consists of two coupled operations: node adaptive rescaling and node adaptive encoder. The idea of node adaptive rescaling is to adaptively learn each node's importance and use it to automatically rescale the output signal of the original graph convolutional operator; In addition to this customized rescaling, each node can also adaptively choose its encoder function according to its importance, forming the concept of node adaptive encoder. With these two techniques in neighborhood aggregation, we present the diagonal graph convolutional networks (DiagGCN), which is able to dynamically depict and model nodes' importance in GCNs.

Importantly, there are several advantages of the proposed DiagGCN model. First, efficiency wise, the adaptive rescaling is achieved by a simple diagonal matrix operation, making DiagGCN's complexity the same as the sparse version of GCNs, i.e., linear with the network volume. In addition, the rescaling step is also parallelizable across all nodes in the graph. Second, it naturally supports both transductive and inductive learning, meaning that the model can generalize inference to unseen nodes. Third, DiagGCN can be connected with graph attention with different mechanisms, including node attention, (multi-hop) edge attention, and path attention, providing a new perspective to understand graph attention. This flexible framework potentially offers a principled way to automatically choose good attention scheme, or reasonable permutation on the adjacency matrix, rescaling diagonal matrices, and adaptive encoders (i.e., AutoML for graph convolution/attention). Specifically, we show the GAT model (Velickovic et al., 2018) can be reformulated as a special case of the proposed DiagGCN model with edge attention. Finally, our extensive experiments on several graph benchmark datasets of different types, including Cora, Citeseer, Pubmed citation networks, and the protein-protein interaction network (PPI) show that the proposed framework consistently achieves performance superiority over existing state-of-art GCN based baselines.

## 2 DEFINITIONS AND PRELIMINARIES

Let $G = (V, E, X)$ denote a graph, where $V$ is a set of $|V| = n$ nodes, $E$ is a set of $|E| = m$ edges between nodes, and $X \in R^{n \times d}$ is a feature matrix recording $d$ features associated with each node. Further let $A \in \{0, 1\}^{n \times n}$ denote the adjacency matrix of the graph $G$, with an element $A_{ij} > 0$ indicating node $v_i \in V$ has a link to $v_j \in V$.

Our node representation framework is built upon graph convolutional networks. Here, we refer to the definition of graph convolution in spectral methods and it can be fast approximated by the multi-layer graph convolution networks (GCNs) (Kipf & Welling, 2016) with the following layer-wise propagation rule:

$$H' = \sigma(\hat{A}HW) \tag{1}$$

where $\hat{A} = \widetilde{D}^{-\frac{1}{2}} \widetilde{A} \widetilde{D}^{-\frac{1}{2}}$ is the normalized adjacency matrix, $\widetilde{A} = A + I_n$ is the adjacency matrix with augmented self-connections, $I_n$ is the identity matrix, $\widetilde{D}$ is the diagonal degree matrix with $\widetilde{D}_{ii} = \sum_j \widetilde{A}_{ij}$, and $W$ is a layer-specific learnable weight matrix. Function $\sigma(.)$ denotes a nonlinear activation function. $H \in R^{n \times D}$ is the matrix of activations, or the $D$-dimensional hidden node representation in the corresponding layer with $H^{(0)} = X$.

## 3 THE DIAGONAL GRAPH CONVOLUTIONAL NETWORKS

In observation of different importance of nodes in real network propagation processes, we present a new graph convolution neural network framework (DiagGCN) with an adaptive neighborhood aggregation rule. The rule is coupled with the node adaptive information rescaling and encoder, where every node can process the graph information distinctly. We also show that the information processing capabilities of different nodes in different propagation steps can be formally implemented with a series of diagonal matrices. Further, sophisticated propagation models (e.g., the graph attention networks with node attention, edge attention, and path attention) can also be formulated as the permutations of the adjacency matrix and diagonal matrices in the DiagGCN model.

### 3.1 BASIC IDEA

In general, our proposed model, DiagGCN, where Diag stands for *diagonal matrix*, is mainly stacked by the following building block:

$$H_i^{'} = \sigma_i[(P\hat{A}QHW)_i] \tag{2}$$

In each layer, compared with the original GCNs, we only add a diagonal matrix $P$ and a diagonal matrix $Q$ to the left and right sides of the normalized adjacency matrix $\hat{A}$, respectively. Importantly, all $P_{ii}$, $Q_{ii}$, and the node-wise activation function $\sigma_i$ are *adaptively* adjusted according to the input information on node $i$, and their concrete forms will be introduced in Sections 3.2 and 3.3.

**Intuition.** Complex propagation or diffusion processes over a network are driven by the diversity of nodes' status and influence (Kempe et al., 2003). Nodes play different roles and process the input information signal in different ways. For example, the structural hole nodes may amplify the information across different groups of nodes in social contagion (Burt, 2009), and opinion leader nodes are more likely to have a broad spread of information to the public, thus may amplify most kinds of their information (Katz, 1957).

However, in most existing GCN models (Kipf & Welling, 2016; Hamilton et al., 2017; Chen et al., 2018), the neighborhood aggregation step can be considered as a simple diffusion process over networks, by ignoring the difference between nodes. To this end, we propose to bridge the gap between GCNs' simple neighborhood aggregation and real-world network diffusion. The proposed Diag-GCN model realizes different ways of processing information on different nodes through the node adaptive rescaling factor and encoder functions. In other words, each node has an adaptive information gate, which is also simpler than sophisticated GCN models with parameterized adjacency matrices, such as MoNet (Monti et al., 2017) and GAT (Velickovic et al., 2018). Furthermore, the permutations of the adjacency matrix and diagonal matrices in DiagGCN offer a general and flexible framework to design powerful GCNs.

Computationally, DiagGCN adds the diagonal matrices to the original GCN architecture, thus the computational complexity is also linear with the volume of the network, which is the same as the sparse version of GCNs. Meanwhile, the computation of node adaptive rescaling factors can be parallelized across all nodes.

### 3.2 NODE ADAPTIVE RESCALING

In Eq. 2, we use the diagonal matrix $Q$ to rescale the rows of the signal $HW$ from the left. In particular, the signal $(HW)_i$ on node $i$ is rescaled by the factor $Q_{ii}$ adaptively adjusted by node $i$.

Figure 1 illustrates the rescaling process. Basically, we use a learnable vector $\vec{s}_1$ to denote the current network state, and the importance of node $i$ is the dot product similarity of its signal $(HW)_i$ and the network state $\vec{s}_1$, that is,

$$r_i = \vec{s}_1^T (HW)_i \tag{3}$$

and the rescaling factor is the sigmoid function (or other monotonic like softplus and exponential function) of the node importance, i.e.,

$$Q_{ii} = \text{sigmoid}(r_i) \tag{4}$$

Dropout mechanism is also applied on $Q_{ii}$ to randomly removing the neighbor node in the aggregation step when training. After the rescaling of $Q$ and the propagation of $\hat{A}$, the signal of node

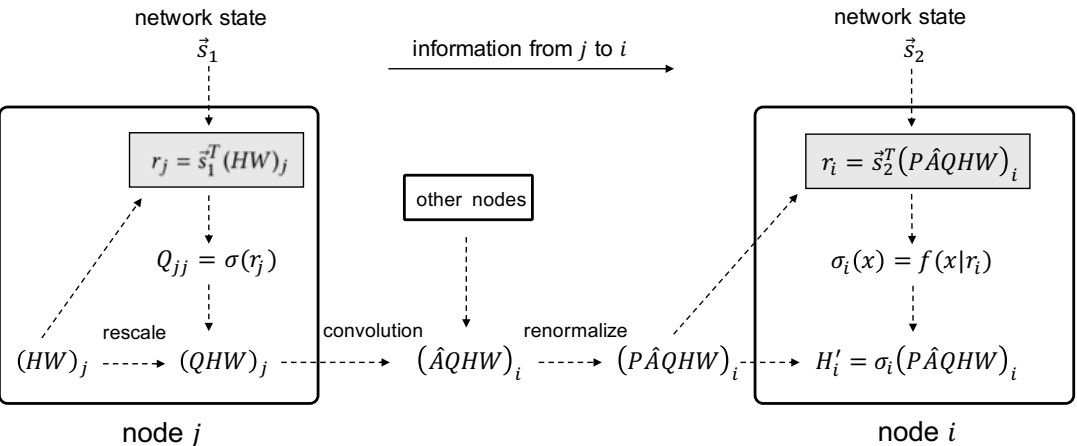

Figure 1: The DiagGCN model. The signal propagation from source node $j$ to the target neighbor $i$ involves $j$'s rescaling function according to its importance, $i$'s convolutional operation which sums over the signals in its neighborhood and again rescales (renormalizes) the resulting signal, and finally $i$ choosing an encoder according to its current importance.

$i$ is transformed as $(\hat{A}QHW)_i$. We then use the diagonal matrix $P$, which is defined below, to renormalize the signals, i.e.,

$$P_{ii} = \frac{1}{\sum_j \hat{A}_{ij} Q_{jj}} \tag{5}$$

Note that $P$ can also have the same rescaling form as $Q$. We will show later how different definitions of $P$ can be characterized as various graph attention mechanisms.

### 3.3 NODE ADAPTIVE ENCODER

At present, the signal of node $i$ is transformed as $(P\hat{A}QHW)_i$. Similarly, we use a new learnable vector $\vec{s}_2$ to represent the new state of the whole network, and the network importance of node $i$ can be then updated as,

$$r_i = \vec{s}_2^T (P\hat{A}QHW)_i \tag{6}$$

Different from previous GCNs (Bruna et al., 2013; Henaff et al., 2015; Defferrard et al., 2016; Kipf & Welling, 2016; Velickovic et al., 2018), we assume that each node can even choose the encoder according to their importance in the network, that is,

$$\sigma_i(x) = \begin{cases} x & , x > 0 \\ \text{sigmoid}(r_i)(e^x - 1) & , x \leq 0 \end{cases} \tag{7}$$

where node importance rescales the activation in the negative axis. The idea is similar to the parameterized ReLu (pReLu) function (He et al., 2015). The difference between pReLu and our setting is that pReLu is shared by the input while our control factor is adaptively adjusted according to the node's importance, which enables the encoder to be a node-wise operation.

### 3.4 MODEL ENHANCEMENTS

We now introduce several possible improvements to DiagGCN.

**Multi-head Propagation Mechanism.** For $i$'s convolutional operation, it first sums over the signals in its neighborhood, and then rescales (renormalizes) the resulting signal adaptively, and finally $i$ chooses an encoder based on its current network importance. Since nodes may have different importance under various situations, we can leverage the multi-head mechanism to further improve the modeling capacity, formally,

$$H_i^{'} = \overset{K}{\underset{k=1}{\Big\|}} \sigma_i^{(k)}[(P^{(k)} \hat{A} Q^{(k)} H W^{(k)})_i] \tag{8}$$

where $\|$ is the concatenation operation, $P^{(k)}$, $Q^{(k)}$, and $W^{(k)}$ are the corresponding matrices in the $k-$th head of the propagation mechanism.

**Multiple-hop Variants.** The propagation layer (Eq. 2) can be generalized to the $k$-hop situation with small modifications, i.e.,

$$H_i^{'} = \sigma_i[(P\hat{A}^k QHW)_i] \tag{9}$$

Compared with Eq. 2, Eq. 9 expands the receptive field to $k$-hop neighborhood. Compared with stacking 1-hop propagation layer $k$ times, Eq. 9 only considers the network importance of the start node and the end node in the propagation path, and omits the intermediates.

We will show later that, when the diagonal matrix $P$ rescales signals in the normalization way, Eq. 9 functions as $k$-hop edge attention mechanism; when the rescaling function of the immediate nodes along the path between two nodes is considered, Eq. 9 functions as $k$-step path attention mechanism.

To summarize, DiagGCN is a flexible framework as the propagation rule can be defined as arbitrary permutations of the adjacency matrices and diagonal matrices. The elements in the diagonal matrices are defined similarly as Eq. 4, rescaling the information signal, or Eq. 5, renormalizing the signal to avoid information vanishing or explosion.

### 3.5 Connection to Graph Attention

Here we discuss DiagGCN's connection with graph attention (e.g., GAT) and show that applying the attention mechanism on nodes, edges and even paths can be regarded as a special matrix permutation case in our DiagGCN model.

Note that when $Q$ and $A$ process signals for different times and $P$ normalizes signals in different range, the model can function as node, edge, and path attention mechanisms. For narration convenience, we temporarily denote the mathematical expression between $P$ and $H$ as $S$. For example, Eq. 2 can be rewritten as $H_i^{'} = \sigma_i[(PSHW)_i]$. We also denote an all-one vector as $\vec{1}$, the diagonal element of $P$ and $Q$ forms the vector $\vec{p}$ and $\vec{q}$ respectively. The division in this subsection is an element-wise operation.

**Node Attention.** When $P$ normalizes the output signals over the global node set, the propagation mechanism functions as the node attention mechanism. $S$ and $\vec{p}$ are expressed below,

$$S = \hat{A}Q, \quad \vec{p}_i = P_{ii} = \frac{1}{\vec{q}^T \vec{1}} \tag{10}$$

**Edge Attention.** When $P$ normalizes the output signals over its neighbors, the propagation mechanism functions as edge attention, i.e.,

$$S = \hat{A}Q, \quad \vec{p} = \frac{1}{S\vec{1}} \tag{11}$$

With this, we show that the GAT model (Velickovic et al., 2018) can be considered as a special implementation of edge attention in our model. Note that the coefficient of edge attention in GAT can be rewritten as

$$\alpha_{ij} = \frac{\exp\left(\text{LeakyReLU}\left(\vec{s}_0^T W\vec{h}_i + \vec{s}_1^T W\vec{h}_j\right)\right)}{\sum_{k\in\mathcal{N}_i} \exp\left(\text{LeakyReLU}\left(\vec{s}_0^T W\vec{h}_i + \vec{s}_1^T W\vec{h}_k\right)\right)} \tag{12}$$

Since LeakyReLU is monotonic and piece-wise linear, the effect of $\vec{s}_0^T W\vec{h}_i$ can be roughly reduced. Thus GAT is approximately our edge attention instance without the adaptive activation encoder, with $Q_{jj} = \exp(r_j) = \exp(\vec{s}_1^T W\vec{h}_j)$. In other words, GAT implicitly considers the network importance and can be reformulated in our diagonal matrix form, and the edge attention is transformed as node-wise rescaling. In this sense, GAT can be considered as a special case of DiagGCN with edge attention, demonstrating the generalization flexibility of the proposed model.

**$k$-hop Edge Attention.** When $P$ normalizes the output signals over its $k$-hop neighbors, the propagation mechanism functions as the $k$-hop edge attention mechanism,

$$S = \hat{A}^k Q, \quad \vec{p} = \frac{1}{S\vec{1}} \tag{13}$$

Table 1: Dataset Statistics

|  | Cora | Citeseer | Pubmed | PPI |
|---|---|---|---|---|
| Task | Transductive | Transductive | Transductive | Inductive |
| Nodes | 2,708 (1 graph) | 3,327 (1 graph) | 19,717 (1 graph) | 56,944 (24 graphs) |
| Edges | 5,429 | 4,732 | 44,338 | 818,716 |
| Classes | 7 | 6 | 3 | 121 (multilabel) |
| Features | 1,433 | 3,703 | 500 | 50 |
| Traning Nodes | 140 | 120 | 60 | 44,906 (20 graphs) |
| Validation Nodes | 500 | 500 | 500 | 6,514 (2 graphs) |
| Test Nodes | 1,000 | 1,000 | 1,000 | 5,524 (2 graphs) |
| Label Rate | 0.052 | 0.036 | 0.003 | 0.789 |

$k$**-hop Path Attention.** When $P$ normalizes the output signals which have been rescaled and convoluted multiple times by the nodes along the path, the propagation mechanism functions as the path attention mechanism. The $k$-step path attention mechanism is expressed in our framework below,

$$S = \hat{A}Q_k\hat{A}Q_{k-1}...\hat{A}Q_1, \quad \vec{p} = \frac{1}{S\vec{1}} \tag{14}$$

After normalization of $P$, the signal $(PSHW)_i$ of node $i$ is fed into our node-wise adaptive encoder $\sigma_i$. The $k$-hop edge attention and path attention mechanisms are generalizations of Eq. 9.

## 4 EXPERIMENTS

We follow the common procedure (Velickovic et al., 2018; Kipf & Welling, 2016; Hamilton et al., 2017) to conduct both the transductive learning and inductive learning experiments on open benchmark datasets and baselines. The experimental results suggest the strong expressive power of the simple DiagGCN framework.

### 4.1 EXPERIMENTAL SETUP

**Benchmark Datasets.** We conduct the transductive experiments on three benchmark citation networks: Cora, Citeseer, and Pubmed, and the inductive experiments on the PPI (protein-protein interaction) dataset. Table 1 summarizes the dataset statistics. The Cora, Citeseer, & Pubmed datasets consist of papers as nodes and citation links as directed edges. Each node has a human annotated topic as the corresponding label and content-based features. The semi-supervised learning task on these citation networks is to predict the topic of each paper, given the content features and the citation relationships to other papers. *For fair comparison, we follow exactly the same data splits as Yang et al. (2016), Kipf & Welling (2016), and Velickovic et al. (2018), with 20 nodes per class for training, 500 overall nodes for validation, and 1000 nodes for evaluation.* The Protein-Protein Interaction dataset (PPI), as processed and described by Hamilton et al. (2017), consists of 24 disjoint subgraphs, each corresponding to a different human tissue. 20 of those subgraphs are used for training, 2 for validation, and 2 for testing, as partitioned by Hamilton et al. (2017).

**Baselines.** For transductive experiments, We compare against both traditional methods without graph convolutions as well as state-of-the-art graph convolution baselines. The non-GCN baselines include Multilayer Perceptron (MLP), manifold regularization (ManiReg) (Belkin et al., 2006), semi-supervised embedding (SemiEmb) (Weston et al., 2012), label propagation (LP) (Zhu et al., 2003), DeepWalk (Perozzi et al., 2014), Iterative Classification Algorithm (ICA) (Lu & Getoor, 2003), and Planetoid (Yang et al., 2016).

We consider the following graph convolution networks as strong baselines: **Chebyshev** (Defferrard et al., 2016) designs fast localized convolutional filters on graph via Chebyshev polynomial expansion. **GCN** (Kipf & Welling, 2016) uses an efficient layer-wise propagation rule that is based on a first-order approximation of spectral convolutions on graphs. **MoNet** (Monti et al., 2017) parametrizes the edge weight in graph and defines the convolution as weighted sum in neighborhood. **DPFCNN** (Monti et al., 2018) generalizes MoNet and defines the convolution operations both on the graph and its dual graph. **GAT** (Velickovic et al., 2018) uses self-attention mecha-

Table 2: Summary of classification accuracy (transductive learning) (%).

| Category | Method | Cora | Citeseer | Pubmed |
|---|---|---|---|---|
| Non-GCN Methods | MLP | 55.1 | 46.5 | 71.4 |
| | ManiReg | 59.5 | 60.1 | 70.7 |
| | SemiEmb | 59.0 | 59.6 | 71.1 |
| | LP | 68.0 | 45.3 | 63.0 |
| | DeepWalk | 67.2 | 43.2 | 65.3 |
| | ICA | 75.1 | 69.1 | 73.9 |
| | Planetoid | 75.7 | 64.7 | 77.2 |
| Graph Convolution | Chebyshev | 81.2 | 69.8 | 74.4 |
| | GCN | 81.5 | 70.3 | 79.0 |
| | MoNet | 81.7±0.5 | – | 78.8±0.3 |
| | DPFCNN | 83.3±0.5 | 72.6±0.8 | – |
| | GAT | 83.0±0.7 | 72.5±0.7 | 79.0±0.3 |
| | GAT-16* | 83.3±0.6 | 72.4±0.7 | 78.8±0.3 |
| | DiagGCN (Rescaling alone) | 83.5±0.6 | 72.9±0.7 | 79.1±0.3 |
| | DiagGCN (Encoder alone) | 83.5±0.6 | 72.9±0.8 | 79.1±0.4 |
| | DiagGCN (1/2) | 83.6±0.7 | 72.9±0.7 | 79.0±0.5 |
| | DiagGCN (1/4) | 83.3±0.7 | 72.5±0.9 | 78.9±0.5 |
| | DiagGCN (1/8) | 82.4±0.9 | 71.5±1.2 | 78.9±0.6 |
| | DiagGCN | **83.6±0.6** | **73.2±0.6** | **79.1±0.3** |
| $p$-value (DiagGCN vs. GAT-16*) | | 5e-4 | 1e-15 | 3e-11 |

Table 3: Summary of classification results in terms of F1 (inductive learning) (%).

| Method | PPI |
|---|---|
| Random | 39.6 |
| MLP | 42.2 |
| GraphSAGE-GCN | 50.0 |
| GraphSAGE-mean | 59.8 |
| GraphSAGE-LSTM | 61.2 |
| GraphSAGE-pool | 60.0 |
| GraphSAGE | 76.8 |
| GAT | 97.3 ±0.2 |
| DiagGCN (Rescaling alone) | 98.4±0.2 |
| DiagGCN (Encoder alone) | 97.2±0.2 |
| DiagGCN (1/2) | 98.4±0.2 |
| DiagGCN (1/4) | 97.9±0.3 |
| DiagGCN (1/8) | 97.4±0.3 |
| DiagGCN | **98.5 ±0.1** |
| $p$-value (DiagGCN vs. GAT) | 2e-12 |

nism to parametrize the edge weight and do the convolution via the weighted sum in neighborhood. Finally, **GAT-16** sets the number of attention heads in the first layer of GAT to 16.

For inductive learning experiments, where test data is unseen during training, GAT and several GraphSAGE variants are considered as baselines.

**Parameters** On Cora, Citeseer, and Pubmed datasets, we employ a two-layer DiagGCN. The architecture and hyper-parameters of models on Cora and Citeseer are exactly the same. The learning rate is 0.005 and l2 weight decay is 0.0005. The first layer consists of 16 propagation mechanisms with 1-hop and 2-hop propagation layers; the hidden dimension is 8; and the last layer only uses 1 propagation mechanism. Dropout mechanism with drop rate 0.6 is applied to the elements of the first layer diagonal matrix, and the adjacency matrix and inputs of each layer. On Pubmed, the learning rate is 0.01 and l2 weight decay is 0.001, the first layer consists of 16 propagation mechanisms with only 1-hop propagation layers, and the hidden dimension is 8. The output of the last layer is averaged over 8 propagation mechanisms.

On PPI, a three-layer DiagGCN is employed, with four propagation mechanism and 256 hidden dimension in the first and the second layers, and the output of the last layer is averaged by six propagation mechanisms.

## 4.2 EXPERIMENTAL RESULTS.

Following the procedure in the original GCN and GAT papers, the reported accuracy results of our methods are averaged over *100 runs* with random weight initializations. In addition, we also report the standard deviations and statistical test results.

Table 2 summarizes the prediction accuracy of node classification for the transductive learning task. It shows that our DiagGCN significantly outperforms GCN and its variants, MoNet, DPFCNN, GAT and GAT with 16 attention mechanism. Thus the explicit consideration of network importance of nodes and adaptive neighborhood aggregation step in GCNs can perform better than other sophisticated models due to its capacity to capture the propagation process in networks.

The results of inductive learning in PPI are shown in Table 3, where the results of different versions of GraphSAGE in the original paper (Hamilton et al., 2017) and the result of GraphSAGE modified by Velickovic et al. (2018) are all reported. GAT (Velickovic et al., 2018) is also added as a baseline. The table suggests that DiagGCN significantly outperforms state-of-the-art baselines—various GraphSAGE models and the latest GAT model.

**Ablation Study.** In the ablation experiments, the results show that DiagGCN with only node adaptive rescaling or with only node adaptive encoders still performs better than baselines.

To further evaluate the importance of the rescaling matrices and encoders in the proposed DiagGCN model, we reduced #heads (#rescaling diagonal matrices and encoders) to 1/2, 1/4, and 1/8 of the original, and found that the performance gradually decreases. DiagGCN (1/4) is comparable to GAT (16 heads). This indicates that adding rescaling matrices/encoders into DiagGCN is very helpful. On the other hand, we can also observe that the model is still robust. Even removing (or keeping only) 1/2 of rescaling matrices and encoders causes a little loss of performance. In addition, it also suggests that the performance of using rescaling or encoder alone is comparable to the model with 1/2 of rescaling matrices and encoders, indicating that rescaling matrices and adaptive encoders play a similar role.

**Statistical Significance.** We did the significance test for the comparison between DiagGCN and GAT. All reported results are averaged over **100** repeated experiments with $p$-values consistently and significantly $\ll 0.05$, which verifies the significance of the proposed model. Additionally, the standard deviation reported indicates the robustness of the results.

## 5 RELATED WORK

With the advances in deep learning, there have been tremendous attempts to generalize neural networks to graph-structured data. Graph Neural Networks (GNNs) (Gori et al., 2005; Scarselli et al., 2009) are introduced as recursive neural networks, where node features are propagated iteratively until equilibrium. GG-NNs (Li et al., 2015) applies gated recurrent units and modern optimization to GNN to evade the requirement of the convergence to the equilibrium.

The convolutional networks in graph-structured data, differently from the general GNNs, can better capture the inherent graph topology by defining the spatial localized convolutional filter in graph Laplacian spectral space (Bruna et al., 2013; Henaff et al., 2015; Defferrard et al., 2016; Kipf & Welling, 2016) or directly on graph neighborhoods (Duvenaud et al., 2015; Niepert et al., 2016; Monti et al., 2017; Hamilton et al., 2017). Among these graph convolutional methods, graph convolutional networks (GCNs) (Kipf & Welling, 2016) have been widely applied to semi-supervised node classification (Kipf & Welling, 2016), link prediction (van den Berg et al., 2017) and knowledge graphs (Schlichtkrull et al., 2018; Zhang et al., 2019) with promising results.

As for many expressive GCN variants, an intuition behind them is to capture complicated pair-wise dependencies between network nodes. In MoNet (Monti et al., 2017), the node relationship is expressed by the parametric kernel with learnable parameters. DPFCNN (Monti et al., 2018) further generalize MoNet and elaborate the relationship both on the graph and its dual graph. In graph attention network (GAT) (Velickovic et al., 2018), multi-head self-attention mechanism is applied on edges. These methods have achieved the state-of-the-art performance. In this paper, we revisit GCNs' neighborhood aggregation step and enhance the representation capacity by adaptively modeling the propagation mechanism into graph convolutions. The Unsupervised loss of maximizing mutual information in Deep Graph Infomax (Veličković et al., 2019) may further help our aggregation step to encode globally relevant information into the node representation.

## 6 CONCLUSIONS

In this work, we study graph convolutional networks (GCNs) with a focus on neighborhood aggregation. We generate insights into the understanding of GCNs, inspired by which we propose the adaptive neighborhood aggregation technique to advance GCNs. Extensive and standard experiments on benchmark datasets suggest that the presented technique offers GCNs with more expressive power than state-of-the-art GCN variants, such as GraphSAGE, GAT, and Chebyshev, benefiting the downstream graph mining tasks. Importantly, our GCN model offers a flexible framework where the permutations of adjacency matrices and diagonal matrices help to design different kinds of GCNs and it can also be connected with graph attention networks, including node attention, edge attention, and path attention. For future work, we would like to examine the other components in GCNs and broadly, to design more powerful neural architectures for modeling graph data structure.

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
