# OpenReview forum: "Diagonal Graph Convolutional Networks with Adaptive Neighborhood Aggregation"
_ICLR.cc/2020/Conference — Reject_

### Official Review · AnonReviewer3 · 2019-10-11
**Official Blind Review #3**

**Rating:** 3

**Review:**

This paper proposed  a so-called diagonal GCN method with adaptive neighborhood aggregation rules., namely, each node will be associated with an individual importance factor to re-scale the output signal  of the graph convolutional operator, and also an adaptive encoder function is adopted. This is achieved by adding a left diagonal matrix P and a right diagonal matrix Q to the standard message passing of GCN H’ = sigma(AHW), as H’(PAQHW); P and Q are parameterized based on the HW part (node features). Multihead attention and multi-hop versions are also derived. Connection of this formulation to the existing GAT method is also discussed. Empirical results on transductive node classification and inductive node classification are reported against state-of-the-arts.

Overall I feel that the novelty of the proposed method is quite limited. Adding a rescaling on each node seems a minor structural change based on existing GCN framework. Although the authors claim that the GAT method can be deemed as a special case of the proposed methods, the main contribution of inductive attention in GAT should not become a credit of the proposed method. Actually the empirical performance of the proposed method has only a small margin compared with the GAT, on the three node classification benchmark datasets, where the accuracies are around 80%, the improvement of the best version of the proposed methods is only around 0.3%, 0.7%, and 0.1%, respectively. This is somehow predictable since the proposed method does not exploit any extra useful information (such as graph structure) but simply base the new parameters on the existing components of the GCN message passing procedure.

The motivation of the paper is that the diffusion process on the graph should depend on the nodes’ status and influence (for example, some opinion leader nodes should have an amplified spread than other nodes), while existing GCN dies bit examine such ``difference’’ between the nodes. However, the roles or status of the nodes in a network are usually unavailable in practice; and if they can be ever be inferred, one has to use structural clues, Unfortunately, the extra parameters added here, namely the matrix P, Q, are dependent only on the features of the nodes (namely HW part), but not the graph structures,  therefore I do not think such parametrization would lead to significant benefit in improving the diffusion process. Indeed,  the original Gat method (graph attention network) has already achieved adaptive aggregation of the neighborhood information, by learning an inductive attention function  based on the node features; therefore the authors claim that existing GCN “ignores the difference between nodes” sounds very vague, since their new parameters are not capable enough to reflect “node differences” either (as a function of the node features).

Some minor comments
(1)	Figure~1 is not easy to read while it is supposed to visualize the main idea clearly in one picture
(2)	Is there proof that if multi-hop version is used, the results can be improved?
(3)	What is the extra number of parameters compared with the standard GCN message passing？

**Experience Assessment:**

I have published one or two papers in this area.

**Review Assessment: Checking Correctness Of Derivations And Theory:**

I assessed the sensibility of the derivations and theory.

**Review Assessment: Checking Correctness Of Experiments:**

I carefully checked the experiments.

**Review Assessment: Thoroughness In Paper Reading:**

I read the paper thoroughly.

---

### Official Review · AnonReviewer2 · 2019-10-23
**Official Blind Review #2**

**Rating:** 3

**Review:**

This paper is well-written. The authors propose a new graph convolution neural network framework (DiagGCN) with an adaptive neighborhood aggregation step to adaptively scale the output signal for each node. In addition, they also propose an adaptive nonlinear encoder for the node's signal. Interestingly, sophisticated propagation models like graph networks
 with node/edge/path attention can also be formulated as the permutations of the adjacency matrix and diagonal matrices in the DiagGCN.

Generally, from my perspective, the idea in the paper is to reformulate the attention mechanism used in the GCN and this new diagonal mechanism is simpler and more general. One of my concerns is the "node adaptive encoder". Based on the experimental results, I did not see an obvious improvement between DiagGCN and DiagGCN with rescaling alone in Table 2 and Table 3. The classification improvement is 0, 0.1, 0.1, 0.3 with the node adaptive encoder. In addition, when comparing DiagGCN with baseline GAT, which is proposed in 2018, the performance approvement is also limited for the transductive learning tasks.

In conclusion, the idea proposed in the paper is interesting to me but the effectiveness of each module is not very well-supported by the experiments.



**Experience Assessment:**

I do not know much about this area.

**Review Assessment: Checking Correctness Of Derivations And Theory:**

I assessed the sensibility of the derivations and theory.

**Review Assessment: Checking Correctness Of Experiments:**

I carefully checked the experiments.

**Review Assessment: Thoroughness In Paper Reading:**

I read the paper thoroughly.

---

### Official Review · AnonReviewer1 · 2019-10-25
**Official Blind Review #1**

**Rating:** 3

**Review:**

This work proposes node adaptive rescaling and node adaptive encoder to enhance GCN. The proposed method is well explained and simple to implement. The experimental results indicate improvement and use t-test to show it's statistically significant.

Cons:
1. I am not convinced that GCN does not consider node importance. When aggregating information, the degree information is used. In the example of opinion leaders, the nodes corresponding to opinion leaders usually have large degree.
2. The proposed method is very similar to the attention method in "Hierarchical Attention Networks for Document Classification", where the query vector is a learnable vector. The authors should at least cite it and make appropriate discussion.
3. The node adaptive encoder is more suitable to be called a new activation function. Then ablation study on different activation functions should be made.
4. The experiments should cover comparison to more recent works, as well as graph classification tasks. For example, comparison to GIN or extending the proposed method to GIN would be interesting.
5. Examples of how the proposed method learn the node importance correctly should be provided.

Minor Cons:
1. Figure 1 has a part of low resolution.
2. The updated node importance in eqn.(6) should use a modified notation to differ from eqn.(3).
3. The position of Table 2&3 is strange.

**Experience Assessment:**

I have published one or two papers in this area.

**Review Assessment: Checking Correctness Of Derivations And Theory:**

I carefully checked the derivations and theory.

**Review Assessment: Checking Correctness Of Experiments:**

I carefully checked the experiments.

**Review Assessment: Thoroughness In Paper Reading:**

I read the paper at least twice and used my best judgement in assessing the paper.

---

### Decision · Program_Chairs · 2019-12-19

**Decision:**

Reject

**Comment:**

All three reviewers are consistently negative on this paper. Thus a reject is recommended.